# Heterocornols from the Sponge-Derived Fungus *Pestalotiopsis heterocornis* with Anti-Inflammatory Activity

**DOI:** 10.3390/md19110585

**Published:** 2021-10-20

**Authors:** Hui Lei, Xiaoxu Bi, Xiuping Lin, Jianglian She, Xiaowei Luo, Hong Niu, Dan Zhang, Bin Yang

**Affiliations:** 1School of Pharmacy, Southwest Medical University, Luzhou 646000, China; huilei@swmu.edu.cn (H.L.); dawn123454@swmu.edu.cn (H.N.); zhangdan@swmu.edu.cn (D.Z.); 2College of Agriculture and Life Sciences, Kunming University, Kunming 650241, China; bxx2797@163.com; 3CAS Key Laboratory of Tropical Marine Bio-Resources and Ecology, Guangdong Key Laboratory of Marine Materia Medica, South China Sea Institute of Oceanology, Chinese Academy of Sciences, Guangzhou 510301, China; xiupinglin@scsio.ac.cn (X.L.); shejianglian20@mails.ucas.ac.cn (J.S.); 4Institute of Marine Drugs, Guangxi University of Chinese Medicine, Nanning 530200, China; luoxw@gxtcmu.edu.cn

**Keywords:** *Pestalotiopsis heterocornis*, heterocornols, anti-inflammatory activity, sponge-derived fungus

## Abstract

One strain-many compounds (OSMAC) manipulation of the sponge-derived fungus *Pestalotiopsis heterocornis* XWS03F09 resulted in the production of new secondary metabolites. The chemical study of the fermentation, cultivated on 3% artificial sea salt in the rice media, led to the isolation of twelve compounds, including eight new polyketide derivatives, heterocornols Q–X (**1**–**8**), one new ceramide (**9**), and three known analogues (**10**–**12**). The structures and absolute configurations of the new compounds were elucidated by spectroscopic data and calculated ECD analysis. Heterocornols Q (**1**) and R (**2**) are novel 6/5/7/5 tetracyclic polyketide derivatives featuring dihydroisobenzofuran and benzo-fused dioxabicyclo [4.2.1] nonane system, which might be derived from the acetyl-CoA by epoxidation, polyene cyclization, and rearrangement to form the core skeleton. Compound **12** showed moderate or weak antimicrobial activities against with MIC values ranging from 25 to 100 μg/mL. Heterocornols T and X (**7** and **8**) could inhibit the production of LPS-induced NO significantly, comparable to dexamethasone. Further Western blotting analysis showed **7** and **8** markedly suppressed the iNOS protein expression in LPS-induced RAW 264.7 cells in a dose-dependent manner. The result showed that **7** and **8** might serve as potential leads for development of anti-inflammatory activity.

## 1. Introduction

During the past several decades, marine-derived fungi are recognized as an important source of novel drug leads. However, most of their biosynthetic gene clusters are silent under laboratory conditions fungi, whereas only a fraction of gene clusters have been transcribed. In order to enlarge the diversity of metabolites, different strategies such as the one strain-many compounds (OSMAC) strategy [1,2,3,4,5], epigenetic modification [6,7,8], and genome mining have been devoted to activating the silent gene clusters [9,10,11,12,13]. Among them, the OSMAC strategy represents a simple strategy involving the systematic alteration of culture conditions.

Our previous investigation on the marine sponge-derived fungus *Pestalotiopsis heterocornis* XWS03F09 has resulted in the discovery of heterocornols A–P [14,15], pestaloisocoumarins A and B, isopolisin B, and pestalotiol A [16]. For exploring the chemical diversity of microorganisms using the OSMAC strategy, we reinvestigated the secondary metabolites of the strain XWS03F09, with additional 3% artificial sea salt to solid rice medium. Further chemical exploration resulted in the isolation of eight new polyketide derivatives, heterocornols Q–X (**1**–**8**), and one new ceramide (**9**), together with three known analogues, pestalone (**10**) [17], and pestalachlorides A and B (**11** and **12**) [18]. The isolated compounds (**1**–**9** and **12**) were evaluated for the cytotoxic, antimicrobial, and anti-inflammatory activities in vitro. Here, we report the details of the isolation, structure elucidation, and biological activities of these compounds.

## 2. Results and Discussion

The fermentation of the fungus *P. heterocornis* XWS03F09 in the rice media was cultivated for 36 days, made up separately with different culture conditions (0, 1%, 3%, 5% artificial sea salt to solid rice medium), and then was extracted with EtOAc three times. The resulting extracts were analyzed by HPLC. The HPLC-UV profiles of the secondary metabolites indicated that they possessed almost the same metabolites, but with dramatical differences in their major constituents (Appendix A). As the concentration of artificial sea salt in the rice media increased, the strain stopped the production of some metabolites. The strain produced some new metabolites with 3% salinity compared with the previous fermentation. The chemical study of the fermentation, cultivated on 3% artificial sea salt in the rice media, yielded twelve metabolites, including eight new polyketide derivatives, heterocornols Q–X (**1**–**8**), and one new ceramide (**9**) (Figure 1).

Compound **1** was isolated as a white amorphous solid, and its molecular formula was established as C_13_H_14_O_4_ on the basis of HRESIMS at *m*/*z* 257.0783 [M + Na]^+^ (Appendix A). The ^1^H NMR spectrum (Table 1 and Appendix A) of **1** displayed resonances attributed to two aromatic protons at *δ*_H_ 7.01 (d, *J* = 7.9 Hz), and 6.58 (d, *J* = 7.9 Hz) belonging to a 1,2,3,4-tetrasubstituted benzene, one methyl [*δ*_H_ 1.30 (d, *J* = 6.2 Hz), one dioxygenated methine (*δ*_H_ 5.96, s), three oxygenated methines [*δ*_H_ 5.50 (dd, *J* = 8.3, 2.5 Hz), 4.34 (t, *J* = 2.5 Hz), and *δ*_H_ 4.61 (q, *J* = 6.0 Hz)], one oxygenated methylene [*δ*_H_ 5.12 (dd, *J* = 12.1, 2.5 Hz), 4.96 (dd, *J* = 12.1, 2.7 Hz)], one methylene [*δ*_H_ 2.32 (dt, *J* = 13.5, 4.7 Hz), and 1.94 (ddd, *J* = 13.5, 11.3, 2.8 Hz)]. The ^13^C NMR and DEPT spectra (Appendix A) of **1** displayed 13 carbon signals corresponding to one methyl, two methylenes, six methines including two olefinic carbons, four oxygenated carbons, and four quaternary carbons. The NMR features of **1** were very similar to those of vaccinol P [19], indicating that **1** was a polyketide derivative structurally related to vaccinol P. The major difference was the presence of an eight-membered carbon ring in **1**, instead of the C-8 a side chain in vaccinol P, which was supported by the additional HMBC correlations from H-13 (*δ*_H_ 5.96) to C-11 (*δ*_C_ 81.4), C-10 (*δ*_C_ 77.3), C-5 (*δ*_C_ 127.1), C-6 (*δ*_C_ 129.4), and C-7 (*δ*_C_ 143.7), from H-11 (*δ*_H_ 4.34) to C-13 (*δ*_C_ 106.4), C-14 (*δ*_C_ 21.1), together with the COSY correlations H-8/H-9/H-10/H-11/H-14 (Figure 2, Appendix A ). Furthermore, the HMBC correlations between H-1 (*δ*_H_ 4.96/5.12) and C-3 (*δ*_C_ 153.4), C-7 (*δ*_C_ 143.7), C-8 (*δ*_C_ 82.6), and C-2 (*δ*_C_ 127.4) indicated that a five-membered carbon ring was fused with ring B through C-2 and C-7. The HMBC correlations from H-11 (*δ*_H_ 4.34) to C-13 (*δ*_C_ 106.4), C-14 (*δ*_C_ 21.1), from H-9 (*δ*_H_ 2.32/1.94) to C-7 (*δ*_C_ 143.7), C-8 (*δ*_C_ 82.6), and C-10 (*δ*_C_ 77.3) indicated that the C-11 of the vicinal diol side chain was connected with C-13 through an O-bridged hemiketal moiety to form an eight-membered ring, which fused to both rings A and B sharing the same joined carbon C-7 and established a tricyclic 6/5/8 skeleton. Benzene ring A, five-membered ring B, and eight-membered ring accounted for six degrees of unsaturation, and the remaining ones thus required **1** to be tetracyclic. Furthermore, the HMBC correlations between H-10 (*δ*_H_ 4.61) and C-13 (*δ*_C_ 106.4), C-8 (*δ*_C_ 82.6). The different chemical shifts of C-10 (*δ*_C_ 77.3), C-11 (*δ*_C_ 81.4), and C-13 (*δ*_C_ 106.4) indicated that the eight-membered carbon ring was divided into rings C and D. Therefore, **1** was identified to be a 6/5/7/5 tetracyclic polyketide derivative featuring a novel carbon skeleton with a dihydroisobenzofuran and benzo-fused dioxabicyclo [4.2.1] nonane system. The relative configuration of **1** was determined by analyzing the NOESY correlations (Figure 3, Appendix A ). The NOESY spectrum of **1** revealed cross-peaks from H-8 to H-11 and from H-10 to H_3_-14, which, together with the lack of a correlation from H-10 to H-11, led to the determination of the relative configuration of **1** as 8*S*,10*S*,11*S*,13*R*. The ECD calculations of **1** were also carried out to determine the absolute configuration. The result showed that the calculated spectrum of (8*S*,10*S*,11*S*,13*R*)-**1** agreed with the experimental one, indicating the absolute configuration of **1** to be 8*S*,10*S*,11*S*,13*R* (Figure 4). Thus, the structure of **1** was determined and named heterocornol Q (1).

Compound **2** was isolated as a white amorphous solid, and possessed the same molecular formula C_13_H_14_O_4_ as that of **1**, based upon its HRESIMS at *m/z* 233.0806 [M − H]^−^ and ^13^C NMR data (Appendix A). The high resemblances of the 1D and 2D NMR data (Table 1 and Appendix A) of **1** and **2**, indicating that both compounds had the same planar structures [20]. Compared with **1**, the main difference occurred at the configuration at C-8, C-10, C-11, and C-13, which was proved by the carbon chemical shifts of C-8 (Δ*δ*_C_ (−0.1 ppm), C-10 (Δ*δ*_C_ 0.0 ppm), C-11 (Δ*δ*_C_ −4.1 ppm), C-13 (Δ*δ*_C_ −1.1 ppm), and C-14 (Δ*δ*_C_ −8.1 ppm).The NOESY correlations of H-8/H_3_-14 (Figure 3 and Appendix A) along with the NOESY correlations for H-13/H-11 revealed that H-8 and H_3_-14 were positioned on the opposite face relative to H-10 and H-11. On the basis of the above evidence and the presumed biosynthetic pathway, we suggest that the relative configuration of **2** was assigned as 8*R*, 10*R*, 11*S*, 13*S*. This assignment was further confirmed by the calculated ECD spectrum of (8*R*, 10*R*, 11*S*, 13*S*)-**2**, the result of which showed good accordance with the experimental one (Figure 4). Thus, the structure of **2** was determined and named heterocornol R (2).

Compound **3** was isolated as a white amorphous solid, and the molecular formula was determined to be C_21_H_30_O_6_ by HRESIMS, giving a molecular ion peak at *m/z* 401.1955 [M + Na]^+^, indicating seven degrees of unsaturation (Appendix A). The ^1^H NMR spectrum (Appendix A) revealed characteristic signals for two aromatic protons of a 1,2,3,4-tetrasubstituted benzene [*δ*_H_ 6.93 (d, *J* = 8.1 Hz) and 6.63 (d, *J* = 8.1 Hz)], and five methyl groups [*δ*_H_ 2.09 (s), 1.39 (d, *J* = 7.2 Hz), 1.2 (d, *J* = 7.0 Hz), 1.05 (s), and 0.91 (s)]. ^1^H and ^13^C NMR spectroscopic data of **3** (Appendix A) revealed that it possessed the similar structural characteristics with pestalachloride E [21], a chlorinated benzophenone derivative obtained from a marine fungal *pestalotiopsis* sp. The major difference was that one aldehyde group at C-5 and dichloro-substituted phenol at C-3 in pestalachloride E was replaced by one ethoxyethane (C-5) and ethyl ketone side chain (C-3) in **3**, respectively. This was further confirmed by the HMBC correlations from H-1′ (*δ*_H_ 4.60) to C-4 (*δ*_C_ 147.4), C-6 (*δ*_C_ 154.8), C-5 (*δ*_C_ 119.1), and C-3′ (*δ*_C_ 65.5), from H-10 (*δ*_H_ 3.01) to C-3 (*δ*_C_ 40.8), C-2 (*δ*_C_ 55.6), and C-4 (*δ*_C_ 147.4), together with the COSY correlations H-1/H-2/H-3/H-10 and H-7/H-8. HMBC correlations between H-8 (*δ*_H_ 6.93) and C-1 (*δ*_C_ 31.7), C-7 (*δ*_C_ 114.0), and C-4 (*δ*_C_ 147.4) and between H-13 (*δ*_H_ 1.39) and C-11 (*δ*_C_ 208.1), C-12 (*δ*_C_ 74.7) and between H-16 (*δ*_H_ 1.05) and C-14 (*δ*_C_ 72.6), C-2 (*δ*_C_ 55.6), and C-15 (*δ*_C_ 25.3) reconfirmed that the hydroxyl was located at C-14. The planar structure of **3** was confirmed by HSQC, COSY, and HMBC experiments (Figure 2 and Appendix A).

Compound **4** was isolated as a white amorphous solid and had the same molecular formula, C_21_H_30_O_6_, as **3** by HRESIMS data (Appendix A). Detailed analyses of its NMR spectroscopic features implied that **3** and **4** had the same planar structures. Analysis of the 2D NMR spectra confirmed the structure of **4** (Figure 2 and Appendix A).

Unfortunately, in the NOESY spectrum, NOE interactions could not be used to determine the different stereoscopic relationship between **3** and **4**. The apparent difference between **3** and **4** are the opposite Cotton effects (Figure 4) and the opposite optical rotations. To further assign the absolute configuration of **3** and **4**, ECD calculations were performed (Figure 4, Appendix A). According to the literature [22,23], the ECD band between 260 and 285 nm is allied with the ^1^L_b_ transition of the aromatic chromophore. The region of the spectrum from 260 to 285 is diagnostic of the configuration of aromatic compounds. The experimental ECD spectrum for compound **3** displays positive Cotton effects at 278 nm and negative effects at 208 and 300 nm. The calculated ECD spectra for **3** with an *R* configuration at C-12 showed a negative Cotton effect between 280 and 320 nm. The CD spectrum of **4** showed negative (229 and 240 nm) and positive (300 and 208 nm) Cotton effects, indicating that the C-12 of **4** was represented as *S*. As a result, the calculated ECD curve for 2*R*, 3*R*, 12*R*-**3** was the best match with **3**, and 2*R*, 3*S*, 12*S*-**4** was the best match with **4** (Figure 4Appendix A).

The molecular formula of **5** was assigned to be C_14_H_18_O_5_ on the basis of the HRESIMS ion at *m/z* 289.1047 [M + Na]^+^ and ^13^C NMR data (Appendix A). The ^1^H and ^13^C NMR data of **5** (Appendix A) were similar to those of vaccinol O [19]. The major difference was due to the presence of signals of the acetyl moiety (*δ*_H_ 2.04, *δ*_C_ 172.5, and 21.2) in **5**, instead of the carbonyl group in vaccinol O. In addition, signals for an isopentyl unit were absent in the NMR spectra of **5** (Table 2 and Table 3), which was confirmed by COSY and HMBC experiments (Appendix A). In addition, the CD spectrum of **5** exhibited a negative Cotton effect at 204 nm in accordance with that of vaccinol O [19], which allowed assignment of the 3*R* absolute configuration (Figure 4). The coupling constant *J*_H-11/H-12_ = 4.2 Hz revealed that the relative configurations of C-11 and C-12 in **5** were suggested as an *erythro* [24]. In order to discriminate between (11*R*, 12*S*)-**5** and (11*S*, 12*R*)-**5**, the electronic circular dichroism (ECD) spectrum of **5** was calculated and compared with the experimental spectra. As a result, the calculated spectrum of (11*R*, 12*S*)-**5** matched well with the experimental data (Figure 4), indicating absolute configuration of **5**, to be 3*R*, 11*R*, 12*S*.

Compound **6** had the same molecular formula as **5**, which was established by their HRESIMS together with the ^13^C NMR data (Appendix A). Its ^1^H and ^13^C NMR data (Table 2 and Table 3, Appendix A) closely resembled those of **5**, except for an acetyl moiety at C-12 (*δ*_C_ 75.3) in **5** and C-11(*δ*_C_ 76.2) in **6**. Compared with **5**, chemical shift of C-10 (*δ*_C_ 37.1), C-11 (*δ*_C_ 69.7), C-12 (*δ*_C_ 76.2), and C-13 (*δ*_C_ 18.4) of **6** were shifted upfield (Δ3.3 ppm, Δ5.6 ppm) and downfield (Δ5.2 ppm, Δ3.3 ppm), respectively. The planar structure of **6** was confirmed by HSQC, COSY, and HMBC experiments (Figure 2, Appendix A). The shared biogenesis and similar chemical shifts and coupling constants of **6** and **5** suggested the **6** had the same relative configurations for C-3, C-11, and C-12 as those of **5**. In addition, compound **6** showed the similar specific rotation (${\left[\alpha \right]}_{D}^{25}$ −2.5 (*c* 0.1, MeOH)) to that of **5** (${\left[\alpha \right]}_{D}^{25}$ −2.4 (*c* 0.20, MeOH)), which indicated compound **6** had the same configuration. Given that both **5** and **6** were isolated from the same strain, the absolute configuration of **6** was deduced as 3*R*, 11*R*, 12*S*.

The molecular formula of **7** and **8** were assigned as C_21_H_28_O_7_, by analysis of their HRESIMS and ^13^C NMR data (Appendix A). Their NMR spectroscopic features implied that **7** and **8** were a pair of regioisomers. Moreover, the 1D NMR data of **7** and **8** were similar to those of **5** (Appendix A). The only difference was due to the presence of signals of the additional isopentyl unit and acetyl moiety in **8**. Thus, it suggested that **8** was a derivative of **5**, which was confirmed by the HMBC correlation from H-1′ (*δ*_H_ 3.36) to C-4 (*δ*_C_ 141.8), C-6 (*δ*_C_ 130.3), C-5 (*δ*_C_ 126.0), and C-3′ (*δ*_C_ 132.4), from H-5′ (*δ*_H_ 4.69) to C-4′ (*δ*_C_ 21.6), C-2′ (*δ*_C_ 129.7), and 5′-OAc (*δ*_C_ 172.8). Compared with **7**, the only difference was that **7** had an acetyl group at C-11, while **8** had an acetyl group at C-12, which was confirmed by the HMBC correlations of H-12 (*δ*_H_ 4.89) with C-10 (*δ*_C_ 35.0), C-13 (*δ*_C_ 15.0), and 12-OAc (*δ*_C_ 171.4) in **8** (Figure 2, Appendix A). Comparison of the analogous chemical shifts and coupling constants between **5** and **8**, and between **6** and **7**, together with biogenetic considerations based on their coisolation [19,25], we suggest that the absolute configuration of **7** and **8** to have 3*R*, 11*R*, 12*S* configurations, which are the same as those of **5**.

Compound **9**, an amorphous powder, possessed a molecular formula of C_37_H_69_NO_5_ based on its ^13^C NMR data and HRESIMS *m/z* 608.5251 [M + H]^+^ (Appendix A). The ^1^H NMR (Table 2 Appendix A) spectrum of **9** exhibited an amide proton [*δ*_H_ 6.91 (NH, d, *J* = 8.0 Hz)], two disubstituted olefinic protons [*δ*_H_ 5.77 (1H, dt, *J* = 15.4, 6.4 Hz) and *δ*_H_ 5.49 (1H, dt, *J* = 15.4, 6.4 Hz)], a trisubstituted olefinic proton [*δ*_H_ 5.08 (1H, d, *J* = 6.4 Hz)], one oxygenated methylene [*δ*_H_ 4.32 (1H, dt, *J* = 8.1, 4.0 Hz) and *δ*_H_ 4.19 (1H, dt, *J* = 8.1, 3.8 Hz)], two oxygenated methine [*δ*_H_ 4.20 (1H, brs) and *δ*_H_ 4.11 (1H, dt, *J* = 8.1, 3.8 Hz)], aliphatic methylenes (*δ*_H_ 1.24~1.42). Additionally, resonances for three methyl groups, including an olefinic methyl group (*δ*_H_ 1.58, brs) and two terminal methyl groups [*δ*_H_ 0.88 (6H, t, *J* = 7.0 Hz)], were observed. Analysis of the ^13^C NMR data and HSQC spectra (Appendix A) revealed the existence of characteristic signals resonances including an amide carbonyl at *δ*_C_ 173.9, methine carbons at *δ*_C_ 53.0, and aliphatic carbons at *δ*_C_ 22.7~29.7. Detailed analysis of these aforementioned data of **9** suggested it was very similar to those of the previously reported ceramide compound [24], (2*S*, 2′*R*, 3*R*, 4*E*, 8*E*)-N-2′-Hydroxyhexadecanoyl-2-amino-9-methyl-4, 8-octadecadiene-1,3-diol. The major differences are the presence of an additional acetyl group of protons (*δ*_H_ 2.05, 20.9, and 171.3) in **9**, indicating that **9** was a 1-O-acetyl derivative of ceramide compound. Combined with the 2D NMR spectra data, an acetyl group was suggested at C-1 in **9**, which was further confirmed by the HMBC correlation of H-1 (*δ*_H_ 4.32 and 4.19) with C-2 (*δ*_C_ 53.0), C-3 (*δ*_C_ 73.0), and C-1″ (*δ*_C_ 171.3) (Figure 2  Appendix A). Furthermore, based on the *J* values for H-4 (*δ*_H_ 5.49, dd, *J* = 15.4, 6.4 Hz), the geometries of the double bonds for H-4/H-5 was *trans*. Compared to the NMR data from (2*S*, 2′*R*, 3*R*, 4*E*, 8*E*)-N-2′-Hydroxyhexadecanoyl-2-amino-9-methyl-4,8-octadecadiene-1,3-diol [26,27], the chemical shift value of the olefinic methyl group at C-9 (*δ*_C_ 16.0) in compound **9** suggests that the double bond at C-8 is *E* geometry. The length of the fatty acid was determined by the characteristic peaks at *m/z* 253, 296, 314, 336 in the EI-MS (Figure 5  Appendix A) [28]. According to the literature [29] and our NMR data, the chemical shift of C-3 and C-2′ in ceramide compound were almost superimposed with configuration of C-3 and C-2′. Comparing the coupling constant of H-2 and H-3, and ^13^C NMR data of the synthetic (2*S*, 2′*R*, 3*R*, 4*E*, 8*E*)-N-2′-Hydroxyhexadecanoyl- 2-amino-9-methyl-4, 8-octadecadiene-1,3-diol, the configuration of **9** was deduced. Thus, **9** was determined and named pestalotiopsisamide.

Polyketides are common biosynthetic precursor in the process of synthesizing aromatics and macrolides in microorganisms. Compounds **1**–**8** were formed via different polyketide precursors by polyketide synthases. Presumed polyketide precursors are considered to be biogenetically derived from acetyl-CoA by reduction, dehydration, oxygenation, and cyclization to form the core skeleton. Next, presumed polyketide precursors underwent reduction to form a vicinal diol (an intermediate), which could generate a dihydroisobenzofuran through reduction and dehydration. Finally, the dihydroisobenzofuran undergoes dehydration and nucleophilic addition to yielded **1** or **2** (Figure 1), featuring a novel 6/5/7/5 tetracyclic polyketide derivatives.

The isolated compounds (**1**–**9** and **12**) were evaluated for their cytotoxic activities against four human cancer cell lines (Hela, A549, HCT-8, A2780) via MTT assay. However, none of them showed obvious cytotoxicities. They were also tested with their antimicrobial activities against three bacteria and one fungi using a micro broth dilution method (Table 4). Only compound **12** showed antibacterial activities against Gram-positive bacteria *Staphylococcus aureus* and *Bacillus subtilis* and Gram-negative bacteria *Escherichia coli*, with MIC values ranging from 25 to 50 μg/mL. Moreover, **12** also exhibited weak antifungal activities against *Candida albicans* with MIC values 100 μg/mL.

Anti-inflammatory in vitro screening tests activities revealed that **7** and **8** could inhibit the production of LPS-induced NO in RAW 264.7 cells significantly (Figure 6A,B) with no cytotoxicity, comparable to the positive drug dexamethasone (DXM, 33 μM). Further Western blotting analysis showed **7** and **8** markedly suppressed the iNOS protein expression in LPS-induced RAW 264.7 cells in a concentration-dependent manner (Figure 6C,D). The result showed that the two new polyketide derivatives, heterocornols T and X (**7** and **8**), might serve as potential leads for development of anti-inflammatory activity.

## 3. Materials and Methods

### 3.1. General Experimental Procedures

Optical rotations were determined using an AntonPaar MCP500 polarimeter. UV spectra were measured with a Shimadzu UV-2600 PC spectrometer. IR spectra were recorded on a Shimadzu IR Affinity-1 spectrometer with KBr pellets. 1D and 2D NMR spectra were collected on a Bruker DRX-500 spectrometer, *δ* in ppm rel. to TMS, *J* in Hz. HRESIMS were performed using a Bruker maXis TOF-Q mass spectrometer (Bruker, Daltonics, Billerica, MA, USA). Silica gel (100–200 mesh, 200–300 mesh, Qingdao Marine Chemical Ltd., Qingdao, China), Sephadex LH-20 (GE Healthcare Bio-sciences AB, Uppsala, Sweden), YMC*GEL ODS-A (S-50 μm, 12 nm) (YMC Co., Ltd., Kyoto, Japan) were used for column chromatography. Semipreparative HPLC was performed using an ODS column (YMC-ODS-A, 250 × 10 mm, 5 μm). CD spectra were measured on a Biologic MOS-450 spectra polarimeter (Biologic Science, Claix, France). ECD spectra were measured with a Chirascan circular dichroism spectrometer (Applied Photophysics). MTT and antimicrobial assays were analyzed using a microplate reader (BioTek Synergy H1, BioTek Instruments, Inc., Winooski, VT, USA).

### 3.2. Fungal Material

The fungal strain XWS03F09 was isolated from the sponge *Phakellia fusca*, which was collected from the Xisha Islands of China in 2012. It was identified as *Pestalotiopsis heterocornis* XWS03F09 by analysis of its ITS region of the rDNA as described, which has been deposited in the GenBank database (accession no. JN943628.1). A voucher specimen (No. XWS03F09) was deposited in the School of Pharmacy, Southwest Medical University, Luzhou, Sichuan, China.

### 3.3. Fermentation, Extraction, and Isolation

The fungal strain *Pestalotiopsis heterocornis* XWS03F09 was grown at 28 °C without shaking for 36 days in 1000 mL conical flasks containing solid rice medium (each flask contained 200 g of rice, 6 g of artificial sea salt; 200 mL of distilled water, boiled in an autoclave for 20 min at 121 °C). The total of fermented rice cultures was extracted with EtOAc three times to afford 165 g of crude extract (Appendix A).

The extract was subjected to silica gel column chromatography eluting with a gradient CH_2_Cl_2_-MeOH (30:1–0:100) to give 9 fractions based on TLC properties. Fraction 3 was separated by silica gel column chromatography eluting with petroleum ether-EtOAc (6:1) to give seven subfractions (Frs. 3.1–3.7). Fr. 3.2 was subjected to Sephadex LH-20 chromatography (MeOH) to afford three subfractions (Frs. 3.2.1–3.2.3). Frs. 3.2.2 was purified by semipreparative HPLC (70% MeOH/H_2_O) to afford **1** (7.0 mg) and **2** (3.0 mg). Fr. 3.3 was subjected to Sephadex LH-20 chromatography to produce four subfractions (Frs. 3.3.1–3.3.4). Fr. 3.3.2 was separated by ODS column chromatography eluting with MeOH-H_2_O (60%) to yield **5** (7.0 mg). Fr. 3.4 was subjected to a Sephadex LH-20 column (MeOH) and purified by preparative TLC to give **9** (4.0 mg). Fraction 4 was isolated by column chromatography on silica gel eluting with petroleum ether-EtOAc (4:1–0:1) to yield five subfractions (Frs. 4.1–4.5). Fr. 4.1 was separated by semipreparative HPLC (60% MeOH/H_2_O) to produce **3** (3.5 mg) and **4** (8.0 mg). Fr. 4.2 was separated by a Sephadex LH-20 column eluting with MeOH to yield three subfractions (Frs. 4.2.1–4.2.3). Fr. 4.2.3 was separated by repeated column chromatography and further purification with semipreparative HPLC (60% MeOH/H_2_O) to give **10** (2.0 mg), **11** (1.0 mg). Fr. 4.3 was subjected to silica gel column chromatography eluting with a CH_2_Cl_2_-Acetone to yield five subfractions (Frs. 4.3.1–4.3.5). Fr. 4.3.3 was purified by semipreparative HPLC (55% MeOH/H_2_O) to afford **12** (7.0 mg). Fraction 5 was subjected to silica gel column chromatography eluting with CH_2_Cl_2_-Acetone (3:1) to yield five subfractions (Frs. 5.1–5.5). Fr. 5.3 was separated over an ODS column eluting with a gradient of MeOH and H_2_O and further purified by semipreparative HPLC with MeOH-H_2_O (40:60) to afford **7** (4.0 mg) and **8** (4.0 mg). Fr. 5.3 was fractionated on Sephadex LH-20 (MeOH) to provide four subfractions (Fr. 5.3.1–5.3.4). Fr. 5.3.2 was repeatedly purified using semipreparative HPLC with MeOH-H_2_O to give **6** (20.0 mg).

*Heterocornol Q (**1**)*: white amorphous solid; ${\left[\alpha \right]}_{D}^{25}$ − 5.1 (*c* 0.4, MeOH); UV (MeOH) *λ*_max_(log *ε*) 280 (2.64), 271 (2.66), 226 (3.33), 208 (3.74) nm; CD (MeOH) λ_max_ (Δ*ε*) 206 (−3.19), 235 (0.39), IR (film)*ν*_max_ 3360, 2916, 1612, 1550, 1373, 1290, 1199, 1105, 1053, 927, cm^−1^; ^1^H NMR and ^13^C NMR data, see Table 1; HRESIMS *m/z* 257.0783 [M + Na]^+^ (calculated for C_13_H_14_NaO_4_, 257.0790).

*Heterocornol R (**2**)*: white amorphous solid; ${\left[\alpha \right]}_{D}^{25}$ +46.6 (*c* 0.1, MeOH); UV (MeOH) *λ*_max_(log *ε*) 280 (3.66), 271 (2.67), 223 (4.32), 212 (4.47) nm; CD (MeOH) λ_max_ (Δ*ε*) 207 (5.68), 236 (−1.57), IR (film)*ν*_max_ 3340, 2916, 1612, 1506, 1377, 1290, 1197, 1161, 1080, 1051, 1014, 927 cm^−1^; ^1^H NMR and ^13^C NMR data, see Table 1; HRESIMS at *m/z* 233.0806 [M − H] ^−^ (calculated for C_13_H_13_O_4_, 233.0814).

*Heterocornol**S (**3**)*: white amorphous solid; ${\left[\alpha \right]}_{D}^{25}$ −1.4 (*c* 0.2, MeOH); UV (MeOH) *λ*_max_(log *ε*) 287 (3.36), 206 (4.18) nm; CD (MeOH) λ_max_ (Δ*ε*) 208 (−1.41), 278 (0.45), 300 (−0.12), IR (film)*ν*_max_ 3331, 3298, 2972, 2930, 2852, 1717, 1636, 1456, 1373, 1240, 1076, 1018, 667 cm^−1^; ^1^H NMR and ^13^C NMR data, see Table 1; HRESIMS *m/z* 401.1955 [M + Na]^+^ (calculated for C_21_H_30_NaO_6_, 401.1940).

*Heterocornol T (**4**)*: white amorphous solid; ${\left[\alpha \right]}_{D}^{25}$ +1.9 (*c* 0.2, MeOH); UV (MeOH) *λ*_max_(log *ε*) 287 (3.43), 206 (4.21) nm; CD (MeOH) λ_max_ (Δ*ε*) 208 (2.25), 240 (−0.24), 279 (−0.50), 299 (0.96), IR (film)*ν*_max_ 3323, 3244, 2968, 2862, 1717, 1647, 1603, 1456, 1373, 1240, 1188, 1151, 1081, 1041, 815 cm^−1^; ^1^H NMR and ^13^C NMR data, see Table 1; HRESIMS *m/z* 401.1937 [M+Na]^+^ (calculated for C_21_H_30_NaO_6_, 401.1940).

*Heterocornol U (**5**)*: white amorphous solid; ${\left[\alpha \right]}_{D}^{25}$ −2.4 (*c* 0.20, MeOH); UV (MeOH) *λ*_max_(log *ε*) 276 (3.58), 269 (3.59), 218 (4.35) nm; CD (300 μg/mL); CD (MeOH) λ_max_ (Δ*ε*) 200 (−2.85), 250 (0.08), IR (film)*ν*_max_ 3404, 3335, 2938, 1713, 1599, 1470, 1373, 1254, 1151, 1022, 1007, 835 cm^−1^; ^1^H NMR and ^13^C NMR data, see Table 2 and Table 3; HRESIMS *m/z* 289.1047 [M + Na]^+^ (calculated for C_14_H_18_NaO_5_, 289.1052)

*Heterocornol**V (**6**)*: white amorphous solid; ${\left[\alpha \right]}_{D}^{25}$ −2.5 (*c* 0.1, MeOH); UV (MeOH) *λ*_max_(log *ε*) 276 (3.72), 269 (3.73), 220 (4.51) nm; IR (film)*ν*_max_ 3414, 2972, 2934, 1732, 1653, 1602, 1498, 1456, 1373, 1240, 1028 cm^−1^; ^1^H NMR and ^13^C NMR data, see Table 2 and Table 3; HRESIMS *m/z* 289.1046 [M + Na]^+^ (calculated for C_14_H_18_NaO_5_, 289.1052).

*Heterocornols**W (**7**)*: white amorphous solid; ${\left[\alpha \right]}_{D}^{25}$ −10.0 (*c* 0.2, MeOH); UV (MeOH) *λ*_max_(log *ε*) 273.8 (3.62), 213 (4.40) nm; IR (film)*ν*_max_ 3414, 2940, 2870, 1732, 1717, 1640, 1456, 1373, 1149, 1024 cm^−1^; ^1^H NMR and ^13^C NMR data, see Table 2 and Table 3; HRESIMS *m/z* 415.1755 [M + Na]^+^ (calculated for C_21_H_28_NaO_7_, 415.1733).

*Heterocornol**X (**8**)*: white amorphous solid; ${\left[\alpha \right]}_{D}^{25}$ −14.6 (*c* 0.1, MeOH); UV (MeOH) *λ*_max_(log *ε*) 272 (3.78), 208 (4.60) nm; IR (film)*ν*_max_ 3414, 2941, 2870, 1732, 1716, 1650, 1490, 1485, 1373, 1240, 1026 cm^−1^; ^1^H NMR and ^13^C NMR data, see Table 2 and Table 3; HRESIMS *m/z* 415.1745 [M + Na]^+^ (calculated for C_21_H_28_NaO_7_, 415.1733).

*(2S, 2′R, 3R, 4E, 8E)-N-2′-Hydroxyhexadecanoyl-2-amino-9-methyl-4, 8-octadecadiene-yl acetate (**9**)*: amorphous powder; ${\left[\alpha \right]}_{D}^{25}$ +13.0 (*c* 0.1 MeOH); ^1^H NMR and ^13^C NMR data, see Table 2 and Table 3; HRESIMS *m/z* 608.5251 [M + H]^+^ (calculated for C_37_H_70_NO_5_, 608.5254).

### 3.4. ECD Calculations

The calculations of new compounds were performed by using the TDDFT method as carried out using the Gaussian 09 program [30]. Conformational analysis was initially conducted by using the Spartan’14 software. Conformers with a Boltzmann distribution over 5% (the relative energy within 6 kcal/mol) were chosen for ECD calculations at the B3LYP/6-311+G(d,p) level. The ECD spectra of different conformers were generated using the program SpecDis by applying a Gaussian band shape with a 0.30 eV width, according to the Boltzmann-calculated contribution after UV correction.

### 3.5. Cell Culture and Cytotoxicity Assay

The hela, A549, HCT-8, A2780, and RAW 264.7 cells were cultured in DMEM medium with 10% FBS, 2 mM glutamine, 100 U/mL of penicillin, and 100 μg/mL of streptomycin at 37 °C under 5% CO_2_ atmosphere. The cytotoxicities of **1**–**9** and **12** against five cell lines, including hela, A549, HCT-8, A2780, and RAW 264.7, were evaluated by the MTT method described in the literature [31].

### 3.6. Antimicrobial Assay

A micro broth dilution assay as previously reported [32] was used to evaluate the MICs of **1**–**9** and **12** against three bacteria (*Staphylococcus aureus* ATCC 25923, *Bacillus subtilis* ATCC 6633, *Escherichia coli* ATCC 25922), and one fungi (*Candida albicans* MYA-2867). The MIC was defined as the lowest concentration of the antimicrobial agent that completely inhibited visual growth of an organism. Ciprofloxacin and amphotericin B (Sigma Inc.) were used as positive controls against bacteria and fungi, respectively.

### 3.7. Determination of Nitric Oxide Production

The RAW 264.7 cells seeded at 1 × 10^6^ cell/well in six-well plates. After 24 h, the cells pretreated with compounds **7** (0, 3, 11, 33 μM), **8** (0, 3, 11, 33 μM), and dexamethasone (DXM, 33 μM) for 2 h, and then stimulated with LPS (1 μg/mL) for 24 h. The supernatant was collected for detecting the NO production with the Griess method.

### 3.8. Western Blot Analysis

The RAW 264.7 cells were pretreated with compounds **7** (0, 3, 11, 33 μM), **8** (0, 3, 11, 33 μM), and DXM (33 μM) for 2 h, and then stimulated with LPS (1 μg/mL) for 24 h. Total protein was extracted via RIPA (Beyotime, Beijing, China) and the concentration measured by BCA protein assay kit (Beyotime, Beijing, China). The (40 μg) protein was separated with 10% SDS-PAGE and transferred onto PVDF membrane (Millipore, Billerca, MA, USA), which was blocked for 1 h with 5% non-fat milk in TBS at room temperature. Then, the membrane was incubated with primary antibody overnight, then washed three times with TBST and incubated with horseradish peroxidase conjugated secondary antibody for 1 h at room temperature, washed three times with TBST and visualized by CEL (Millipore). GAPDH served as an internal control. Band pattern was analyzed with Fluor Chem FC3 system (ProteinSimple, San Francisco, CA, USA).

### 3.9. Data Analysis

The data were expressed as the mean ± S.E. of at least three independent experiments. The statistical significance of the differences between the means was determined either using Student’s t-test or one-way analysis of variance where appropriate. If the means were found to be significantly different, multiple pairwise comparisons were carried out by Tukey’s post hoc test. The threshold value for acceptance of difference was 5% (*p* ≤ 0.05).

## 4. Conclusions

Eight new polyketide derivatives, heterocornols Q–X (**1**–**8**), one new ceramide (**9**), and three known analogues (**10**–**12**) were isolated from the sponge-derived fungus *Pestalotiopsis heterocornis* XWS03F09. The structures and absolute configurations of the new compounds were elucidated by spectroscopic data and calculated by ECD analysis. Compound **12** displayed growth inhibition towards *S. aureus*, *B. subtilis*, and *E. coli*, with MIC values ranging from 25 to 100 μg/mL. Heterocornols T and X (**7** and **8**) could inhibit the production of LPS-induced NO significantly, comparable to dexamethasone. The result showed that **7** and **8** might serve as potential leads for development of anti-inflammatory activity.

## Data Availability

The data presented in this study are available in the main text and the Appendix A of this article.

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
