# Peer review of "Heterocornols from the Sponge-Derived Fungus Pestalotiopsis heterocornis with Anti-Inflammatory Activity"

_marinedrugs, 2021, doi:10.3390/md19110585_

Round 1

Reviewer 1 Report

The authors report on the isolation of new and known metabolites from the sponge-derived fungus P. heterocoronis, and their biological evaluation in which only two compounds show anti-inflammatory activity worthy of interest.

The structural elucidation of the molecules has been defined by extensive NMR analysis and the absolute configuration established by comparison between experimental and simulated CD spectra.

However, at page 9, line 230, the assignment of C8=C9 (E)stereochemistry similarly to the C4=C5 double bond is not possible, because the latter one is a disubstituted alkene showing the expected J value, which cannot be present in the other trisubstituted bond. Therefore, the authors must apply another method to establish the stereochemistry of this bond, e.g. by the NMR values for the methyl group in comaprison with reported data.

In the biosynthetic pathway proposed in Scheme 1, hemiketalization is not correct because hemiketals are formed by nucleophilic addition of an alcohol on a carbonyl group, and not between two OH groups.

Author Response

Q1:The authors report on the isolation of new and known metabolites from the sponge-derived fungus P. heterocoronis, and their biological evaluation in which only two compounds show anti-inflammatory activity worthy of interest.

Reply: Thank you very much for your comments.

Q2: The structural elucidation of the molecules has been defined by extensive NMR analysis and the absolute configuration established by comparison between experimental and simulated CD spectra.

Reply: Thank you very much for your comments.

Q3: “However, at page 9, line 230, the assignment of C8=C9 (E)stereochemistry similarly to the C4=C5 double bond is not possible, because the latter one is a disubstituted alkene showing the expected J value, which cannot be present in the other trisubstituted bond. Therefore, the authors must apply another method to establish the stereochemistry of this bond, e.g. by the NMR values for the methyl group in comaprison with reported data.”

Reply: We appreciate valuable suggestions and comments from the reviewers.  We have studied reviewer’s comments carefully and have checked the stereochemistry of C4=C5 and C8=C9 again.

We agree with the Reviewer’s comments that the assignment of C8=C9 (E) stereochemistry similarly to the C4=C5 double bond is not possible. The stereochemistry of C4=C5 should has been confirmed by the coupling constant between H-4 and H-5. The other trisubstituted bond should has been confirmed by the NMR values for the methyl group in comaprison with reported data.

The coupling constant of H-4 is correct (H-4, J = 15.4, 6.4 Hz), the geometries of the double bonds for H-4/H-5 was trans. We have removed “Similarly” in the manuscript. “Similarly, the double bond at C-8 of 9 is E geometry” was changed as “Compared to the NMR data from (2S, 2'R, 3R, 4E, 8E)-N-2'-Hydroxyhexadecanoyl-2-amino-9-methyl-4,8-octadecadiene-1,3-diol[1][2], The chemical shift value of the olefinic methyl group at C-9 (δC 16.0) in compound 9 suggests that the double bond at C-8 is E geometry”

Q4: In the biosynthetic pathway proposed in Scheme 1, hemiketalization is not correct because hemiketals are formed by nucleophilic addition of an alcohol on a carbonyl group, and not between two OH groups.

Reply: We agree with this point. We have carefully checked the plausible biosynthetic pathway of compounds 1 and 2 again. Finally, the cyclic ketal in compounds 1 and 2 were formed by dehydration and nucleophilic addition. We have revised our manuscript according to the comments.

Reviewer 2 Report

This manuscript describes the isolation and bioactivity of new heterocornols from the Fungus Pestalotiopsis heterocornis.

The manuscript including structure elucidation, biological activities, tables, figures, and reference is well organized.

It seems to me that the description of absolute configuration of the diastereomers heterocornol Q and heterocornol R is also interesting,

In my opinion, this manuscript could be 'Accept in present form' in 'Marine Drugs'.

Author Response

Q1:In my opinion, this manuscript could be 'Accept in present form' in 'Marine Drugs'.

Reply: Thank you very much for your comments.